

# Direct Lagrangian tracking simulation of droplet growth in vertically developing cloud

Yuichi Kunishima[1] and Ryo Onishi[1]

[1]Center for Earth Information Science and Technology, Japan Agency for Marine-Earth Science and Technology 3173-25 Showa-machi, Kanazawa-ku Yokohama Kanagawa 236-0001 Japan

**Correspondence:** Ryo Onishi (onishi.ryo@jamstec.go.jp)

**Abstract.** We present a direct Lagrangian simulation that computes all the warm-rain processes in a vertically developing cloud, including cloud condensate nuclei (CCN) activation, condensational growth, collisional growth, and droplet gravitational settling. This simulation, which tracks the motion and growth of individual particles, is applied to a kinematic simulation of an extremely-vertically-elongated quasi-one-dimensional domain, after which the results are compared with those obtained
from a spectral-bin model, which adopts the conventional Euler framework. The comparison results, which confirm good bulk statistical agreement between the Lagrangian and conventional spectral-bin simulations, also show that the Lagrangian simulation is free from the numerical diffusion found in the spectral-bin simulation. After analyzing the Lagrangian statistics of the surface raindrops that reach the ground, back-trajectory scrutiny reveals that the Lagrangian statistics of surface raindrops contains the information about the sky where the raindrops grow like the shape of snow crystals does.

## 1 Introduction

Because clouds play central roles in weather and climate, numerous cloud microphysics models have been developed to investigate their physics and predict their development. Conventional models are divided into either bulk or spectral-bin types based on their microphysical representations. In bulk models, all of the microphysical processes, such as mixing ratios or number concentrations of cloud hydrometers (cloud, rain, cloud ice, snow, and graupel, etc.) are described in terms of grid-averaged
parameters. In contrast, in spectral-bin models, hydrometer size or mass distributions are modeled directly. Although spectral-bin models are more complex and prognose more variables than bulk models, both types are the same in the sense that they prognose grid-averaged values. In other words, all conventional models are based on the Euler framework.

However, in recent years, a number of new cloud models that are based on the Lagrangian framework, in which the motion and growth of individual droplets are tracked, have been developed. These Lagrangian cloud models are also divided into
two groups: super-droplet models (e.g., Shima et al., 2009; Riechelmann et al., 2012; Dziekan and Pawlowska, 2017) and direct tracking models (e.g., Onishi et al., 2015; Saito and Gotoh, 2018). The former introduces the multiplicity concept, in which each droplet represents multiple droplets that share approximately the same attributes and positions, while the latter directly computes the motion and growth of each droplet and can consider the influence of microscale flow and scalar field on the droplet motion and growth, even though their computational costs are extremely high. For example, Onishi et al. (2015)




developed the Lagrangian Cloud Simulator (LCS), which adopts the Euler-Lagrangian framework and can provide reference data for cloud microphysical models. Their study revealed the turbulence enhancement of collisional growth on the temporal evolution of droplet size distributions for a periodic box (cubic) domain. Saito and Gotoh (2018) reported on a direct tracking simulation for a periodic box domain that additionally considers condensational growth, while ignoring the cloud condensation nuclei (CCN) activation. Here, it should be noted that other existing direct tracking models still lack the CCN activation process and also rely on periodic box domains, which are useful, but unphysical in the sense that large drops repeatedly reenter the same domain. However, no direct tracking simulations have ever succeeded in representing all of the warm-rain processes, which include CCN activation, condensation/evaporation, collision, and gravitational settling.

With the above background in mind, this study aims to establish a direct Lagrangian model that can compute all of the abovementioned warm-rain processes and obtain Lagrangian statistics on droplet growth in a vertically developing cloud. First, a simple stochastic CCN activation model for direct Lagrangian simulations was developed and implemented in the LCS. The integrated LCS was then applied to a kinematic simulation for a vertically developing warm cloud. The resulting computational model adopts an extremely-vertically-elongated quasi-one-dimensional domain that can allow gravitational settling of large raindrops in physical (rather than periodic) space at a feasible computational cost.

The rest of this paper is organized as follows. In the next section, we describe the numerical methods, including the newly-developed CCN activation model, used for direct Lagrangian simulations. Computational settings, including the domain setting, are described in Sec. 3. Then, results and discussion for both Eulerian and Lagrangian simulations and statistics are presented in Sec. 4, and our conclusions are given in Sec. 5.

## 2 Numerical Methods

### 2.1 Lagrangian Cloud Simulator (LCS) Overview

The LCS (Onishi et al., 2015) adopts a hybrid Euler-Lagrangian framework in which flow motion and scalar transportation are computed using an Euler method, and the particle motion and growth are obtained via a Lagrangian tracking method. Within that framework, cloud microphysics are computed by direct Lagrangian tracking, which follows the motion and growth of individual particles. Previously, the LCS was used primarily to focus on the collisional growth of droplets in a turbulent medium. In this study, in order to include the water droplet phase change, the humidity and temperature fields are calculated using the same Euler method that was used for the flow. In addition, to include the droplet activation (nucleation) process, this study presents a newly developed CCN activation model based on Twomey's modeling (Twomey, 1959). The resulting integrated LCS can track all of the warm-rain processes of water droplets (CCN activation, condensational and collisional growth, and gravitational settling).

The following subsection describes the present kinematic simulation, while Subsection 2.3 describes the present Euler methods used for the calculation of flow and scalar fields. Then, Subsection 2.4 describes the present Lagrangian method used for particle phase, which includes the newly developed CCN activation model designed for use with a Lagrangian framework.





**Table 1.** Initial profiles for the KiD *warm-1* case.

| $z$ [m] | $\Theta$ [K] | $Q_v$ [kg kg$^{-1}$] | $p$ [Pa] | $T$ [K] | $\rho$ [kg m$^{-3}$] |
|---|---|---|---|---|---|
| 3000 | 311.1 | 0.0037 | $6.99 \times 10^4$ | 280.4 | 0.869 |
| 740 | 297.9 | 0.0150 | $9.18 \times 10^4$ | 284.6 | 1.12 |
| 0 | 297.9 | 0.0150 | $1.00 \times 10^5$ | 297.9 | 1.17 |

## 2.2 Overview of quasi-one-dimensional kinematic simulation

### 2.2.1 Kinematic model

5   This study adopts the computational settings of the "Kinematic Driver" (KiD) model (Shipway and Hill, 2012), which was originally designed to provide an inter-comparison framework for bulk and spectral-bin cloud microphysics models. The flow and temperature profiles are prescribed in a way that minimizes feedback between dynamics and microphysics, and thus facilitate straightforward comparisons. Here, we adopt the simplest test, which is a shallow convection case (*warm-1* case) in which a simple updraft is prescribed as

$$ w(t) = \begin{cases} w_0 \sin(\pi t/600) & \text{for } t < 600\,\text{s} \\ 0 & \text{otherwise} \end{cases}, \tag{1} $$

with $w_0 = 2\,\text{m s}^{-1}$. The duration and depth of the simulation are 3600 s and 3000 m, respectively. Table 1 shows the initial temperature and moisture profiles, which are set to be similar to those used in the Global Energy and Water Cycle Experiment (GEWEX) Cloud System Study (GCSS) Rain in Cumulus over the Ocean (RICO) composite intercomparison. Inside the surface layer below 740 m, the air is assumed to be well mixed and neutrally stratified. More specifically, the potential temperature
15   $\Theta (= T(p/p_0)^{R/C_p}$, where $T$ is the temperature, $p$ is the pressure, $R$ is the gas constant of dry air ($R$=287 J kg$^{-1}$K$^{-1}$), $C_p$ is the specific heat at constant pressure ($C_p$=1004 J kg$^{-1}$K$^{-1}$) and $p_0$(=$1.00 \times 10^5$ Pa) is the reference pressure) and the initial vapor mixing ratio $Q_v$ are set to the same values inside the layer.

### 2.2.2 MSSG-Bin simulation (a spectral-bin simulation)

The Multi-Scale Simulator for the Geoenvironment (MSSG; Takahashi et al. (2013); Matsuda et al. (2018)) has a warm-
20   bin/cold-bulk hybrid cloud microphysics model named MSSG-Bin (Onishi and Takahashi, 2012) in which the mass coordinate $m$ was discretized as $m_k = 2^{1/s} m_{k-1}$, and where $s$ was set to 16. The representative radius of the first bin was 2.7 $\mu$m and 528 classes were calculated, the largest class of which had a representative radius of 5.4 mm. The vertical grid spacing was set to $\Delta_z$=25 m and the time interval was set to $\Delta t$=1 s. Following the KiD protocol, the initial CCN concentration was set to $5.0 \times 10^7$ m$^{-3}$ and kept constant for the entire simulation.



### 2.3 Flow and scalar phase in the Euler framework

In addition to solving the three-dimensional (3D) continuity and Navier-Stokes equations for incompressible flows based on a finite difference method, the LCS solves the transport equations of scalars $\phi$ as

$$\left\{\frac{\partial}{\partial t}+\boldsymbol{U}\cdot\boldsymbol{\nabla}\right\}\phi=\kappa_\phi\boldsymbol{\nabla}^2\phi+S_\phi, \tag{2}$$

where $\boldsymbol{U}$ is the flow velocity, $\kappa_\phi$ is the diffusion coefficient and $S_\phi$ is the source term. The LCS considers two scalars; the $\phi$ scalar, which can be either the water vapor mixing ratio $Q_v$, or the potential temperature $\Theta$.

The spatial derivatives were calculated using fourth-order central differences. The conservative scheme devised by Morinishi et al. (1998) was used for the advection terms and the second-order Runge-Kutta scheme was used for time integration. Since the temperature field for the KiD *warm-1* simulation is fixed, only the transport equation for $Q_v$ was calculated explicitly in this study. The diffusion coefficient for the water vapor is $\kappa_q = \nu/\mathrm{Sc}$, where $\nu$ is the kinematic viscosity and Sc is the Schmidt number (Sc=0.675). The vapor field was coupled with the particle phase through the source term $S_q$, described in Eq. (12) in

Subsection 2.5.

### 2.4 Particle phase in the Lagrangian framework

#### 2.4.1 Motion of particles

Under the limit of a large ratio of the density of the particle material to that of the fluid ($\rho_p/\rho \gg 1$, where $\rho_p$ and $\rho$ are the densities of particle and air, respectively), the governing equation for the $i$-th particle is given by

$$\frac{\mathrm{d}\boldsymbol{V}_{p,i}}{\mathrm{d}t}=-\frac{f_i}{\tau_{p,i}}\left[\boldsymbol{V}_{p,i}-\left(\boldsymbol{U}(\boldsymbol{x}_{p,i},t)+\boldsymbol{u}(\boldsymbol{x}_{p,i},t)\right)\right]+\boldsymbol{F}_{\mathrm{impulse},i}+\boldsymbol{g}, \tag{3}$$

where $\boldsymbol{V}_p$ is the particle velocity, $\boldsymbol{U}(\boldsymbol{x}_p,t)$ is the air velocity at the particle position, $\boldsymbol{u}$ is the disturbance flow velocity caused by the surrounding particles, and $\tau_p$ is the particle relaxation time defined as $\tau_p = (2/9)(\rho_p/\mu)r^2$, where $\rho_p$ is the particle density ($\rho_p$=1,000 kg m$^{-3}$), $\mu$ the air viscosity ($\mu$=1.79×10$^{-5}$ Pas at the standard atmosphere) and $r$ is the particle radius. The impulsive acceleration $\boldsymbol{F}_{\mathrm{impulse}}$ represents the force resulting from collisions and $\boldsymbol{g}$ is the gravitational acceleration vector [$\boldsymbol{g}=(0,0,-g)$

with $g=9.81$ m s$^{-2}$]. The non-linear drag coefficient $f$ (Rowe and Henwood, 1961) is obtained by $f = 1 + 0.15\,(\mathrm{Re}_p)^{0.687}$, where the particle Reynolds number $\mathrm{Re}_p$ is defined by $\mathrm{Re}_p = (|\boldsymbol{U}-\boldsymbol{V}_p|\,2r_p)\,/\,\nu$.

   One important physical process that is often neglected because of the high computational cost associated with its simulation is hydrodynamic interaction (HI) between moving particles. While moving in a flow medium, a particle induces a flow disturbance (denoted by $\boldsymbol{u}$ in Eq.(3)) in its vicinity that may reduce collisions between approaching particle pairs. The significance of HI on the collisional growth of cloud droplets with O(10$\mu$m) sizes is well known and has actually been confirmed recently by a Lagrangian tracking simulation (Onishi et al., 2015). This study considers HI using the binary-based superposition method (BiSM) (Onishi et al., 2013), which is one of the so-called superposition methods, and which considers HI based on a





pair-wise technique while assuming that interactions via three or more particles are negligible. This assumption dramatically

reduces computational costs while maintaining reliability as long as the particle number concentration is small, as is found in

atmospheric clouds. It should be noted that the superposition method, which assumes Stokes flows around moving particles,

is valid for small particles whose $\mathrm{Re}_p$ is smaller than unity. This requirement typically corresponds to $r < 40\mu$m. For larger

droplets, associated errors will grow but the influence of HI will become insignificant as the particle inertia increases. As a

result, the significance of HI relative to particle inertia would level off at a certain particle size, the threshold of which would

depend on the flow conditions. While a more precise discussion on such issues could be done with size-resolving simulations,

they are not a central focus of this study. Furthermore, the lubrication effect is not included in the superposition method. This

would cause a slight overestimation of the collisional growth rate in the present simulations. Again, while this issue could be

solved by cutting-edge size-resolving simulations, such detailed discussions are out of the scope of this study.

The second-order Runge-Kutta method was used for the time integration. The flow velocity $U$ at the droplet position was

linearly interpolated from the adjacent grid values. Since the $\Delta t$ time interval should be smaller than the relaxation time

, it should be set to very small value if a tiny particle is included in the domain. In order to avoid extremely small time

intervals, we designate particles whose relaxation times are smaller than the time interval as tracer particles. In this study,

we set the time interval to $1.25 \times 10^{-3}$ s and designate particles with a radii smaller than 14 $\mu$m as tracers. This treatment

does not significantly alter the growth of particles with radii smaller than the threshold because condensational growth is more

dominant than collisional growth. Note that all the CCN (dry aerosol) particles, whose sizes were not treated explicitly, were

also considered as tracers.

### 2.4.2 Growth of particles

The growth of the $i$-th liquid droplet is calculated as

$$\frac{\mathrm{d}m_{p,i}}{\mathrm{d}t} = \left(\frac{\mathrm{d}m_{p,i}}{\mathrm{d}t}\right)_{\mathrm{act}} + \left(\frac{\mathrm{d}m_{p,i}}{\mathrm{d}t}\right)_{\mathrm{cond}} + \left(\frac{\mathrm{d}m_{p,i}}{\mathrm{d}t}\right)_{\mathrm{coll}}, \tag{4}$$

where the subscripts 'act', 'cond' and 'coll' represent the growth due to activation, condensation and collision, respectively. It

should be noted that 'cond' represents the change due to phase change and includes evaporation shrinkage.

**CCN activation**

Next, we explain our newly developed stochastic CCN activation method for direct Lagrangian tracking simulations. First, the

number density of newly-activated liquid droplets at the corresponding time step is diagnosed in each Euler grid cell (i.e., inside

a $\Delta_x \times \Delta_y \times \Delta_z$ volume, where $\Delta$ is the grid spacing and the subscript denotes the direction) using the bulk air saturation and

bulk number liquid droplet density. Then, a stochastic judgment is made on whether or not to activate each CCN particle. If the

decision is affirmative, another stochastic judgment is then made on which size is to be set.

The formulation of the liquid droplet activation process is based on the relationship between the number of activated CCN

$N_{\mathrm{act}}$ and the supersaturation ratio $S_w = (Q_v - Q_{v_{\mathrm{sw}}})/Q_{v_{\mathrm{sw}}}$, where $Q_{v_{\mathrm{sw}}}$ is the saturated mixing ratio with respect to water.



The saturated mixing ratio is obtained as $Q_{v\,sw} = \varepsilon e_s/(p - e_s)$, where $\varepsilon$ is the ratio of the gas constants for dry air and water

vapor (i.e., $\varepsilon = R/R_v = 0.622$) and $e_s$ is the saturated vapor pressure given by the Tetens formula $e_s = 611.2 \exp\{17.67(T - 273.15)/(T - 29.65)\}$ (Murray, 1967). In Twomey (1959), the relationship between $N_{\mathrm{act}}$ and $S_w$ is written in the form $N_{\mathrm{act}} = CS_w^k$, where $C$ and $k$ depend on the CCN type. If we define $S_{\max}$ as the supersaturation needed to activate the total particle count $N_{\mathrm{CCN}} + N_w$, where $N_{\mathrm{CCN}}$ and $N_w$ are the number concentrations of dry CCN and liquid droplets, then $C$ can then be represented as $C = (N_{\mathrm{CCN}} + N_w)S_{\max}^{-k}$. Thus, the number of activated CCN (tiny liquid droplets) can be expressed as

$$N_{\mathrm{act}} = (N_{\mathrm{CCN}} + N_w)\left(\frac{S_w}{S_{\max}}\right)^k. \tag{5}$$

The number of newly-activated droplets is calculated as

$$N_{\mathrm{act}} = \max\left\{0, (N_{\mathrm{CCN}} + N_w)\min\left[1, \left(\frac{S_w}{S_{\max}}\right)^k\right] - N_w\right\}. \tag{6}$$

Assuming the maritime conditions given in (Onishi and Takahashi, 2012), the parameters $k$ and $S_{\max}$ were set to 0.6 and 0.008, respectively. The conversion from a dry aerosol to a liquid droplet is stochastically determined using the probability $N_{\mathrm{act}}/N_{\mathrm{CCN}}$ for each dry aerosol particle. The activated (nucleated) liquid droplet size ($r_{\mathrm{act}}$) is then determined stochastically so that its size distribution follows the exponential probability distribution given in Soong (1974) as

$$f(r_{\mathrm{act}}) = \frac{3r_{\mathrm{act}}^2}{\bar{r}^3}\exp\left[-\left(\frac{r_{\mathrm{act}}}{\bar{r}}\right)^3\right], \tag{7}$$

where $\bar{r}$ is the radius of an average mass droplet and it was set to 11 $\mu$m (Onishi and Takahashi, 2012).

The mass growth due to this activation process is then calculated as

$$\left(\frac{\mathrm{d}m_{p,i}}{\mathrm{d}t}\right)_{\mathrm{act}} = \frac{4\pi\rho_w r_{\mathrm{act},i}^3/3}{\Delta t}. \tag{8}$$

**Condensational growth**

The phase change for the $i$-th particle is written as (Houze, 1993)

$$\left(\frac{\mathrm{d}r_{p,i}}{\mathrm{d}t}\right)_{\mathrm{cond}} = \frac{1}{r_{p,i}(F_d + F_k)}\left(S_w - \frac{\alpha/T}{r_{p,i}} + \frac{\beta}{r_{p,i}^3}\right), \tag{9}$$

$$F_d = \frac{\rho_p}{\rho\kappa_q Q_{v\,sw}}, \qquad\qquad F_k = \left(\frac{L_v}{R_v T} - 1\right)\frac{L_v\rho_p}{KT}. \tag{10}$$

Here, $F_d$ represents a thermodynamic term associated with vapor diffusion and $F_k$ is associated with heat conduction. The terms with coefficients $\alpha$ and $\beta$ represent the curvature effect and the reduction in vapor pressure due to CCN hydrophilicity,





respectively. This study simply ignored the solute effect and adopted $\alpha = 3.3 \times 10^{-7}$ m K and $\beta = 0$ m$^3$. $R_v$ is the gas constant

for water vapor ($R_v$=461.7 J kg$^{-1}$K$^{-1}$), $\kappa_q$ is the diffusion coefficient for water vapor, and $K$ is the coefficient of air thermal conductivity. It should be noted that Eq. (9) also covers particle evaporation. When undersaturated (i.e., $S_w < 0$), a liquid droplet shrinks due to evaporation. When a particle becomes smaller than the threshold, it reverts to the CCN (dry aerosol) category. The threshold radius was set to 1 $\mu$m.

The mass growth rate is obtained from

$$\left(\frac{\mathrm{d}m_{p,i}}{\mathrm{d}t}\right)_{\mathrm{cond}} = 4\pi\rho_p r_{p,i}^2 \left(\frac{\mathrm{d}r_{p,i}}{\mathrm{d}t}\right)_{\mathrm{cond}} . \tag{11}$$

**Collisional growth**

It was assumed that colliding particles are immediately united by conserving mass and momentum. In other words, the coalescence efficiency was unity, and subsequent breakups (collisional breakups) were not considered. This no-breakup assumption can be justified for shallow clouds like the present KiD *warm-1* case, and the fact that none of the other currently used spectral-

bin models actually consider breakups for such cases. A collision is judged from the trajectories of a droplet pair by assuming linear particle movement for the time interval $\Delta t$.

## 2.5   Phase coupling

This study adopts one-way momentum coupling, thereby neglecting the momentum feedback from the particle phase to the flow phase. Since the mass fraction of the particles to flow is O($10^{-3}$) in the present simulations, this is easily justified by the

dilute condition. The vapor and particle phases are coupled in order to conserve the water content. The coupling is done via the source term $S_q$ in Eq. (2).

$$S_q(\boldsymbol{x},t) = -\sum_{i \in \boldsymbol{V}(\boldsymbol{x})} \left\{ \left(\frac{\mathrm{d}m_{p,i}}{\mathrm{d}t}\right)_{\mathrm{act}} + \left(\frac{\mathrm{d}m_{p,i}}{\mathrm{d}t}\right)_{\mathrm{cond}} \right\} \tag{12}$$

# 3   Computational conditions and performance

## 3.1   Quasi-one-dimensional computational domain

Table 2 shows the computational settings for the present LCS simulations. Two different-sized domains were used. This study focuses on the smaller size simulation (*NORM*), which has the horizontal size of ($L_x = L_y =$) 0.01 m and a vertical size of 3000 m. Roughly speaking, the influential length scale of HI relative to the particle diameter is approximately 10 for small Re$_p$ (Ayala et al., 2007; Onishi et al., 2013), although it would be smaller for larger particles. The maximum raindrop diameter is around $10^{-3}$ m in the present simulations. Since the horizontal domain size is 10 times larger than the maximum diameter, the HI of a particle would not affect itself unphysically through the horizontal periodic boundary condition even for the largest drops.





**Table 2.** Computational conditions for the LCS simulations. The same settings are used for both horizontal axes: $x$ and $y$ directions.

| | Domain size $(L_x)^2 \times L_z$ [m³] | Grid spacing $(\Delta_x)^2 \times \Delta_z$ [m³] | Initial number concentration $n_{p0}$ [m⁻³] | Initial total number $N_{p0}$ [-] | Time interval $\Delta t$ [s] |
|---|---|---|---|---|---|
| *NORM* | $(0.01)^2 \times 3000$ | $(3.33 \times 10^{-3})^2 \times (3.91 \times 10^{-2})$ | $5.00 \times 10^7$ | $1.50 \times 10^7$ | $1.25 \times 10^{-3}$ |
| *LARGE* | $(0.03)^2 \times 3000$ | $(3.33 \times 10^{-3})^2 \times (3.91 \times 10^{-2})$ | $5.00 \times 10^7$ | $1.35 \times 10^8$ | $1.25 \times 10^{-3}$ |

The large size simulation (*LARGE*), the volume of which was nine times larger than that of *NORM*, was performed in order to confirm the robustness of the *NORM* simulation as will be discussed in Subsection 4.1. The number of initial particles was set to nine times larger than that for *NORM* in order to keep the particle number concentration the same.

The mean separation length $l_{\text{sep}} \left( = n_{p0}^{-1/3} \right)$ was $2.71 \times 10^{-3}$ m. The horizontal grid spacing $\Delta_x (= L_x/n_x$, where $n_x$ is the number of grid points in the $x$-direction) was set similar to $l_{\text{sep}}$, while the vertical grid spacing $\Delta_z (= L_z/n_z)$ was set longer than the maximum path of the largest drop within a single time interval ($\sim O(10)$ m s⁻¹ $\times O(10^{-3})$ s $\sim O(10^{-2})$ m).

### 3.2  Initial and boundary conditions

The air flow and the temperature field were prescribed as shown in Subsection 2.2.1. The pressure was determined by the hydrostatic equilibrium as

$$\frac{\mathrm{d}p}{\mathrm{d}z} = -\rho g. \tag{13}$$

The obtained pressure and temperature are also listed in Table 1. Air viscosity depends on temperature, and decreases by about 5% from 297.9 K (domain bottom) to 280.4 K (domain top). This study simply neglected this change and used a constant value: $\mu = 1.79 \times 10^{-5}$ Pa s at 298 K and 1 atm. Initially, dry aerosols (CCNs) were randomly and homogeneously distributed. The initial number density was $n_{p0} = 5.00 \times 10^7$ m⁻³, which is the same value used in the MSSG-Bin simulation.

Periodic boundary conditions were applied in all three directions for the particle field, except for the particles that reached the ground surface ($z = 0$). Those particles (hereafter, surface raindrops) are removed from the system immediately after reaching the ground. Due to the prescribed updraft (see Eq. 1) some particles move out from the top boundary and re-enter the domain from the bottom (ground surface). Although this sounds unphysical, it does not alter the physical result since all the reentry particles are dry CCNs (not activated liquid droplets). The prescribed updraft lifts all the air parcels by ($\int w dt =$) 764 m, while there is over 1000 m vertical gap between the domain top and cloud top ($z \sim 2000$ m) as will be seen in Fig. 3. This simple periodic treatment for the vertical direction precisely conserves the number of particles, thus keeping the particle number concentration near the surface, until the first raindrop reaches the surface and disappears.

Periodic boundary conditions were applied in horizontal directions for the vapor field, while the Neumann boundary condition with zero gradient was applied in the vertical direction. This restricts $Q_v$ near the surface to its initial value.



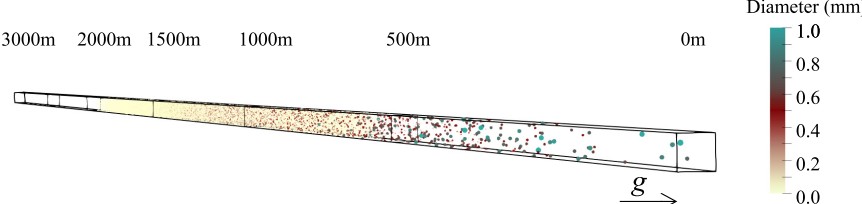

**Figure 1.** Quasi-one-dimensional domain for the direct Lagrangian tracking simulation. Liquid droplets at $t = 1300$ s for a *NORM* simulation are drawn, while the CCN particles (dry aerosols) are not. The color of each droplet denotes its diameter. For more information, see the online Supplementary Movie.

## 3.3 Sensitivity of computational settings and computational performance

We performed 30 production runs for *NORM* and one for *LARGE*. Each run was intense and required massively-parallel high-performance computing. One simulation for *LARGE* required 11 times longer to complete than was required for *NORM*, while the domain size and number of particles for *LARGE* were nine times larger than those for *NORM*. In *NORM*, the ratio of the number of particles to that of grid points was 21.7, which means the particle calculation was much heavier. The particle calculation component actually occupied the 98% of the total computational cost.

Next we checked the effect of the computational settings used for grid spacing and time interval on the results obtained by performing a finer grid resolution simulation for *NORM* with a half-size grid, i.e., $8(=2^3)$ times larger number of grids and comparing the results to the reference setting in Table 2. We also performed an additional *NORM* simulation using time intervals that were finer by a factor 2, i.e., with $\Delta t = 6.25 \times 10^{-4}$ s. The results from those additional simulations were within the statistical fluctuations summarized in Table 4.1, thus indicating the validity of the present computational settings

## 4 Results and Discussion

### 4.1 Bulk statistics

Figure 2 shows the temporal evolution of water paths obtained from the present LCS together with those from the MSSG-Bin. The threshold radius between cloud and rain was set to 40 $\mu$m. All the results from the 30 LCS runs are drawn. However, it should be noted that even though the LCS results look like a single thick line, each LCS run is slightly different from the others as denoted by standard deviations, as shown in Table 3.

The total liquid water path increases due to the supersaturation caused by the prescribed updraft for $t \leq 600$ s, which remains constant after the updraft stops until the first raindrop reaches the ground surface and is removed from the system. Rain water appears at around $t{\sim}600$ s, and increases rapidly until $t{\sim}1350$ s when the raindrops start to reach the surface. These results are consistent between the LCS and MSSG-Bin results. However, there is a difference in the maximum rain water path, which can be attributed to the difference in the CCN treatment. Following the KiD protocol, the MSSG-Bin simulation kept $N_{CCN}$ constant



**Figure 2.** Temporal evolution of liquid water path. The solid lines show the LCS results (all the 30 *NORM* runs are drawn independently while the deviation is small) while the dashed lines show the MSSG-Bin result.

**Table 3.** Comparison of bulk statistical values between *NORM* and *LARGE* runs. The values show (mean)±(standard deviations).

|  | Total amount of surface precipitation [m$^{-1}$] | Total number of surface raindrops [m$^{-2}$] | Maximum rain water path [kg m$^{-2}$] |
|---|---|---|---|
| *NORM* | $(8.60 \pm 0.05) \times 10^{-4}$ | $(1.51 \pm 0.04) \times 10^{7}$ | $1.07 \pm 0.01$ |
| *LARGE* | $8.56 \times 10^{-4}$ | $1.48 \times 10^{7}$ | $1.06$ |

5    while the LCS simulations did not. $N_{\mathrm{CCN}}$ in the LCS simulation decreased with time, since CCN particles were consumed for activation. This led to larger raindrops, and consequently to larger rain water amounts.

Table 3 shows a comparison of the bulk statistical values between the *NORM* and *LARGE* LCS runs. The values for *LARGE* range within the statistical variations of *NORM*. This suggests that the limited size of the *NORM* computational domain did not matter for bulk statistics.

10    Figure 3 shows the temporal evolution of the vertical profile of liquid water content. The vertical spacing of the LCS result for the plot was set to 25 m, which is the same vertical grid spacing used in the MSSG-Bin simulation. The results show that the cloud bottom exists at around $z =600$ m and that the cloud top is at $z=2000$ m. Raindrops appear from the bottom at around $t=1,100$ s. These results are consistent for both the LCS and MSSG-Bin simulations. However, a remarkable difference is seen in the raindrop settling path for $600<t<1,100$ s. The LCS result shows a sharp ridge due to settling inside the cloud layer, while the MSSG-Bin result does not. The MSSG-Bin somehow suffers from numerical diffusion because it adopts the Eulerian approach, while the LCS that uses a Lagrangian tracking approach does not. This clearly shows that the LCS can provide robust diffusionless numerical results.



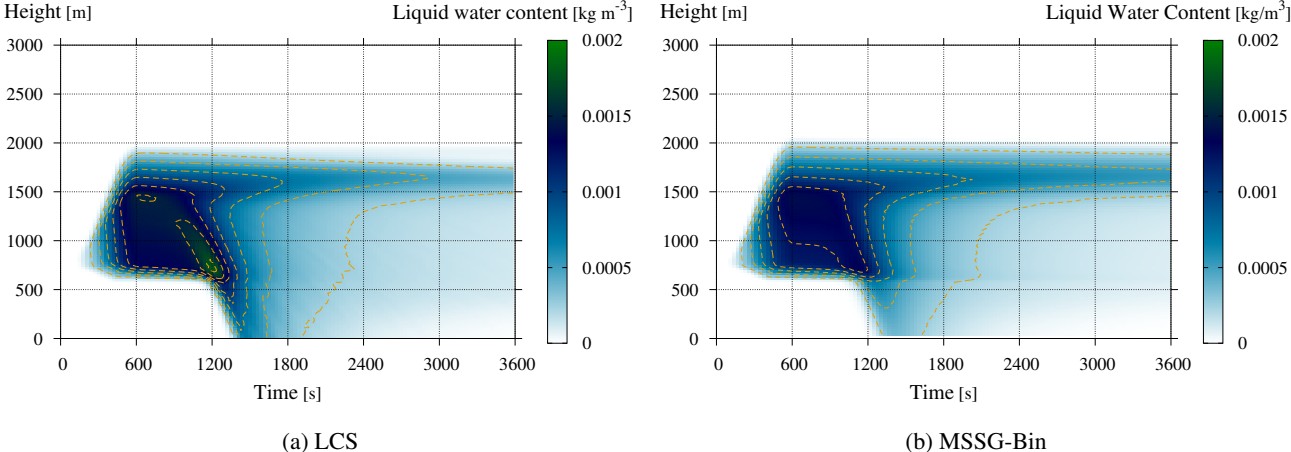

**Figure 3.** Temporal evolution of the vertical profile of liquid water content for (a) LCS and (b) MSSG-Bin. The yellow dashed lines are isolines at every $2.5 \times 10^{-4}\ \mathrm{kg\,m^{-3}}$.

## 4.2  Lagrangian statistics

One of the strengths of direct Lagrangian simulations is that they can provide Lagrangian statistics. This study focuses on the Lagrangian statistics of surface raindrops, which are the raindrops that reach the ground surface ($z = 0$ m). The 30 *NORM* simulations obtained totally 36,018 surface raindrops, which means that each run obtained 1,200 surface raindrops on average.

Figure 4 shows the initial locations of particles that comprise the 36,018 surface raindrops used in the 30 *NORM* runs. The total number of constituent particles was 119,490,655, which means that each surface raindrop contained an average of 3,320 particles. The prescribed updraft lifted the air by 764 m. The particles that were initially below 0 m can be considered in the same manner as those transported into the domain along, or generated near, the surface. This figure shows that surface raindrops consist of the CCN particles initially found below 900 m in height. This kind of information, which cannot be obtained from the conventional Euler-based bulk or spectral-bin simulations, can be used in other studies such as investigations into the chemical compositions of surface raindrops.

Figure 5 plots the surface raindrop volume, $V_{\mathrm{sr}}$, against the number of constituent particles, $N_{\mathrm{memb}}$. The regression line shows $V_{\mathrm{sr}} = 2.38 \times 10^{-14} N_{\mathrm{memb}}$, which means that the average diameter of each constituent is 36.1 $\mu$m. For example, a surface raindrop with a 1 mm diameter, whose volume is $5.24 \times 10^{-10}\ \mathrm{m^{-3}}$, consists of 22,000 constituent particles. Recalling that the mean diameter of the nucleated droplets was set to 22.0 $\mu$m (see Subsection 2.4.2), it can be surmised that condensation caused each constituent to grow from 22.0 to 36.1 $\mu$m, and then to grow further due to collisions. This is consistent with findings that show collisional growth is typically dominant for droplets with diameters larger than 40 $\mu$m.

Figure 6 shows the height against time collision history of the Top-1 surface raindrop. The surface precipitation time $T_{\mathrm{surf}}$ was 1,347 s. Here, we identify the 'mother' particle from the tens of thousands of constituent particles making up the raindrop, each of which has its own ID number in the LCS simulation. When two particles collide and unite, the collector particle grows





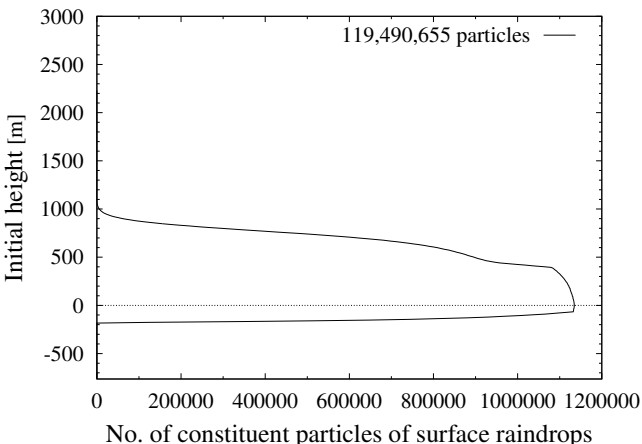

**Figure 4.** Initial locations of the particles that comprise the surface raindrops of the 30 *NORM* runs. The vertical distribution of the 119,490,655 particles that make up the 36,018 surface raindrops is drawn. The particles that are initially at negative height (near the top boundary) can be considered in the same manner as those that are transported into the domain along, or generated near, the surface.

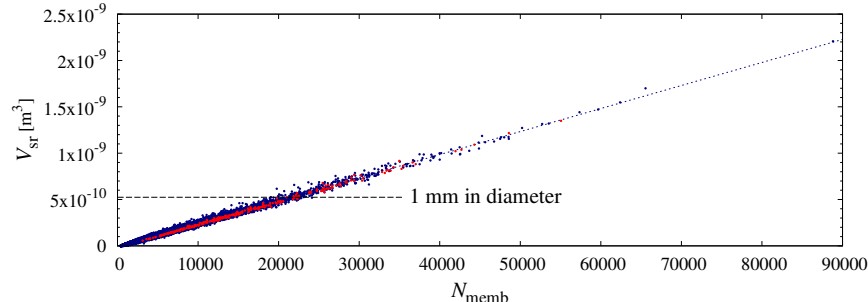

**Figure 5.** Surface raindrop volume, $V_{sr}$, against the number of constituent particles, $N_{memb}$. The regression line shows $V_{sr} = 2.38 \times 10^{-14} N_{memb}$. The Top-10 surface raindrops are denoted by red dots.





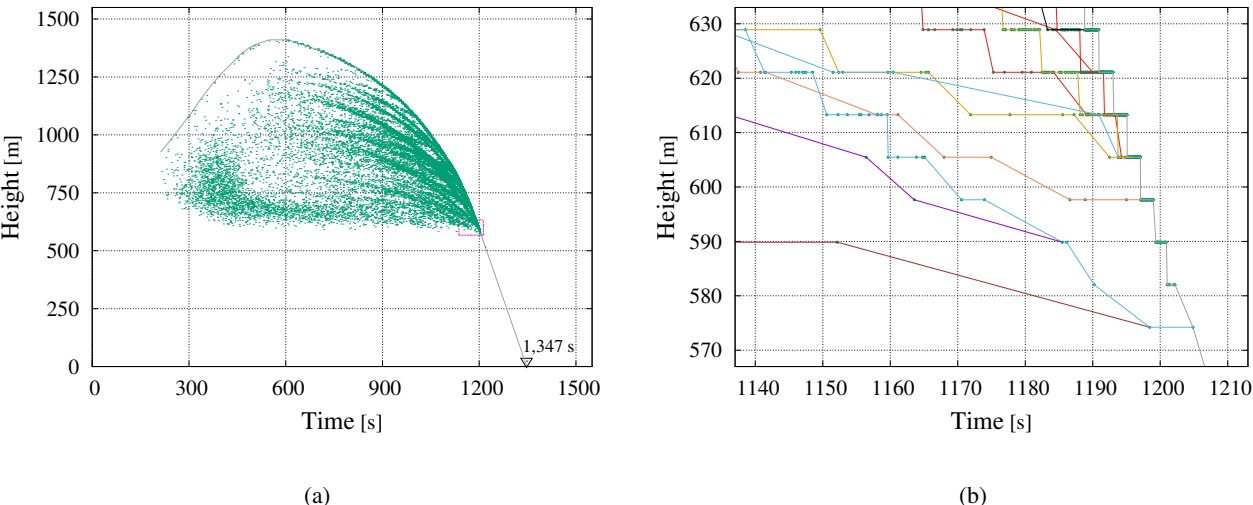

(a)                                                                    (b)

**Figure 6.** Collision events associated with a Top-1 surface raindrop. (a) All the associated collision events are plotted as dots, while the trajectory of the 'mother' particle is shown as a line. (b) The enlarged view of the rectangular window shown in (a). The trajectories of raindrops ($r_p > 40\ \mu\mathrm{m}$) on a vertical resolution with 7.81 m intervals. Each dot represents individual collision event associated with the raindrops shown in colored lines.

5  and the collected particle is removed, thereby conserving mass and momentum. In this process, the particle with the highest attained height of the raindrop constituents survives and its ID number is affixed to the raindrop. That particle is then designated as the 'mother' of the surface raindrop. It should be noted that one particle attained its maximum height at $t \sim 600$ s when the updraft halted. In Fig. 6(a), the trajectory of the mother particle is drawn as a line. The maximum height, $H_{\mathrm{max}}$, attained by the mother particle can be used to characterize the surface raindrop. It should also be noted that the trajectory of a particle with
10  constant settling velocity is drawn in a straight line, whereas the trajectory line of the mother particle appears to be parabolic. This result is attributed to the increasing settling velocity that results from collisional growth.

### 4.3   Top-10 raindrops

Figure 7 shows the relationship between the maximum attained height of the mother particle for each surface raindrop $H_{\mathrm{max}}$ against its surface precipitation time $T_{\mathrm{surf}}$ for all of the 30 *NORM* runs. Totally, 36,018 dots are drawn in the figure. The black
15  dashed line is obtained by least squares regression for all the surface raindrops and is written as

$$H_{\mathrm{max}} = 0.225 T_{\mathrm{surf}} + 1,020. \qquad (14)$$

The positive correlation between $H_{\mathrm{max}}$ and $T_{\mathrm{surf}}$ means that larger $H_{\mathrm{max}}$ leads to larger $T_{\mathrm{surf}}$ because it takes longer for a droplet to fall far to the surface.

Interestingly enough, if restricted to the surface raindrops ranked in the Top-10 with the earliest $T_{\mathrm{surf}}$ for each run, the correlation becomes negative as shown in Fig. 7(b). Although the plots are very scattered, a statistical test has proven that a



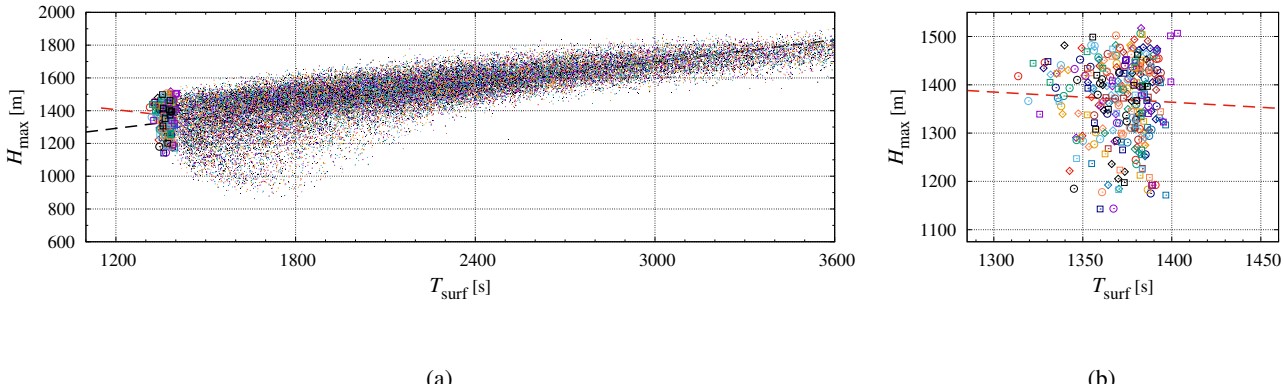

(a)
(b)

**Figure 7.** Maximum attained height of the mother particle for surface raindrops against surface precipitation time, $T_{\mathrm{surf}}$. The circled points represent Top-10 raindrops. Only the plots for the Top-10 raindrops are drawn in (b).

positive correlation is rejected with an 76% confidence level, meaning that a negative correlation is very likely. We have also

verified that the confidence level depends on computational conditions. If the evaporation rate is artificially strengthened by

setting $F_k=0$ in Eq. 9, the confidence level becomes 99% (virtually certain). This adds extra confidence to the supposition that

this correlation reversal is possible under specific conditions.

In order to reveal the correlation reversal mechanism, we additionally collected $T_{600}$, the time when the raindrop passes

$z$=600 m (i.e., near the cloud bottom). Figure 8 plots $H_{\mathrm{max}}$ against $T_{600}$. The black dashed line is obtained by least squares

regression for all the surface raindrops and is written as

$$H_{\mathrm{max}} = 0.296 T_{600} + 987. \tag{15}$$

The sensitivity of $H_{\mathrm{max}}$ to $T_{600}$ is stronger, i.e., the slope is steeper than that for $T_{\mathrm{surf}}$. Raindrops with higher $H_{\mathrm{max}}$ take longer

to fall from $H_{\mathrm{max}}$ to the cloud bottom ($z \sim$600 m). However, they can also collect more droplets during the longer fall and

grow larger, and thus take less time to fall from 600 m to the surface. This weakens the correlation between $T_{600}$ and $T_{\mathrm{surf}}$, and

consequently results in the stronger sensitivity (steeper slope in the linear form) of $H_{\mathrm{max}}$ to $T_{600}$ than to $T_{\mathrm{surf}}$.

In contrast to Fig. 7, the correlation between $H_{\mathrm{max}}$ and $T_{600}$ for the Top-10 is positive in Fig. 8. The sensitivity of $H_{\mathrm{max}}$ to

$T_{600}$ for the Top-10 raindrops is somewhat stronger than that for all the other surface raindrops. That is, the correlation reversal

happens while the Top-10 surface raindrops fall from the cloud bottom to the surface. These facts suggest the plausible scenario

that larger drops among the Top-10 surface raindrops, with higher $H_{\mathrm{max}}$, pass the cloud bottom later and then overtake other

raindrops before they can reach the surface. This kind of overtake would only occur if specific conditions are met.

This scenario can be confirmed in the supplementary movie (see Movie S1). The movie view consists of three windows;

domain side-view (left window), liquid water path (right-top window) and look-up view from the ground (right-bottom win-

dow). The look-up view shows that red spheres (raindrops about 500 $\mu$m in diameter) appear from the cloud bottom at around

$t$=1,100 s, and that the blue spheres (raindrops about 1 mm in diameter) that appear just after tend to overtake the red spheres

before they can reach the ground.





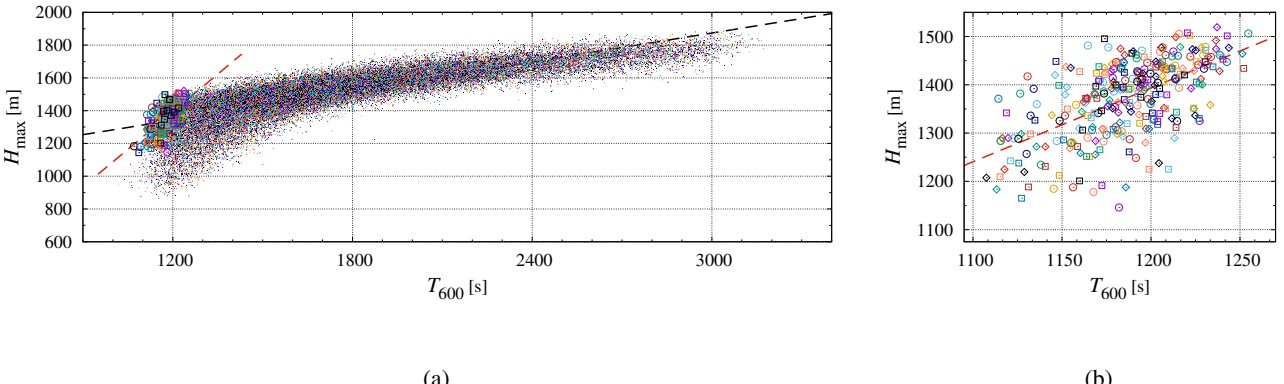

|  (a)  |  (b)  |

**Figure 8.** Attained maximum height against the time when the raindrop passes the height of 600 m (nearly cloud bottom), $T_{600}$. The black and red dashed lines show the least squares regression line for all the surface raindrops and for the Top-10 raindrops, respectively. In (b) only the plots for the Top-10 raindrops are drawn.

The Lagrangian statistics provide information about the surface raindrop growth history. They also show differences for the overall surface raindrops and that for a fraction of earliest raindrops. Another set of simulations show that these differences are sensitive to the evaporation rate. These results suggest that the Lagrangian statistics on the surface raindrops have the potential to provide useful information for estimating atmospheric conditions aloft. They also reminds us of the Nakaya diagram (Nakaya, 1954), which can translate the shapes of snow crystals into information on atmospheric conditions experienced by those crystals as they fell to the surface. The present Lagrangian simulation results clearly show that statistics on raindrops, as well as snow crystals, contain information about the sky that formed them.

## 5  Conclusions

The Lagrangian cloud simulator (LCS; Onishi et al. (2015)) adopts a Euler-Lagrangian hybrid framework, in which the flow motion and scalar transportation are computed using an Euler method, and the particle motion and growth are calculated using a Lagrangian tracking method. In that framework, cloud microphysics are based on direct Lagrangian tracking, which tracks the motion and growth of individual particles. In this study, we developed a CCN activation model for direct Lagrangian tracking simulations and implemented it in the LCS. This implementation enables complete tracking of particle growth from a CCN to surface raindrop, including the CCN activation, condensation/evaporation, collisions, and gravitational settling stages. It should also be noted that HI is considered for colliding droplet pairs in our newly developed LCS, which is currently the only model that can consider HI for tracking the simulation of droplet collisional growth at a reasonable computational cost.

The integrated 3D LCS has previously been applied to a kinematic shallow convection cloud development in the case of the *warm-1* KiD model (Shipway and Hill, 2012), which was originally designed to provide an inter-comparison framework for bulk and bin cloud microphysics models. For the kinematic simulation used in this study, an extremely-vertically-elongated quasi-one-dimensional computational domain has been adopted. Use of this domain enables the representation of vertical





cloud development while keeping the computational domain volume (and consequently the total number of particles to track) small. The unique combination of the high-performance LCS and the extremely-vertically-elongated domain has allowed us, for the first time, to successfully complete a tracking simulation of droplet growth from CCN (dry aerosols) to raindrops in a reasonably realistic manner. In our process, the results obtained from the LCS simulations are first compared to those from the MSSG-Bin model (Onishi and Takahashi, 2012), which is a spectral-bin cloud microphysics model based on the conventional

Euler framework. The comparison results show good agreement, in terms of bulk statistics such as water mixing ratios, between the LCS and MSSG-Bin, except for slight variations due to CCN treatment differences. Furthermore, it has been confirmed that the LCS is free from particle phase numerical diffusion.

One of the strengths of direct Lagrangian simulations is that they can provide Lagrangian statistics. This study has focused on surface raindrops, which are the raindrops that reached the ground surface. For example, a surface raindrop with a 1 mm

diameter consists of approximately 22,000 constituent particles. This means that each constituent particle grew to 36.1 $\mu$m in diameter due to condensation, and then grew to 1 mm in diameter because of collisions. Additionally, by applying back-trajectory analysis to the surface raindrops, it was possible to identify the particle that attained the maximum height among all the constituent particles, which we define as the 'mother' particle for a surface raindrop. These analysis results also show that the maximum attained height of mother particles has a positive correlation on their surface precipitation time. This can be

easily explained by the fact that it takes more time for a particle to fall from higher altitudes. Interestingly, however, if restricted to the Top-10 (the earliest 0.8%) surface raindrops with the earliest surface precipitation time, the correlation was negative. As a plausible mechanism for this, it was suggested that larger drops among the Top-10 surface raindrops, which are lifted up higher, pass the cloud bottom later but overtake the slower falling raindrops before they can reach the surface.

Based on the results obtained in this study, we can conclude that the use of Lagrangian statistics for evaluating surface rain-

drops can provide useful information for estimating atmospheric conditions aloft. These results also remind us of the Nakaya diagram (Nakaya, 1954), which can interpret the shapes of snow crystals into information on the atmospheric conditions experienced by the crystals. Since our present Lagrangian simulation clearly shows that the statistics of surface raindrops, as well as snow crystals, contain information the conditions aloft, additional discussion becomes possible, thereby indicating that direct Lagrangian models can be powerful and promising meteorological tools.

*Acknowledgements.* This research was supported by the Japanese Ministry of Education, Culture, Sports Science and Technology (MEXT) as an "Exploratory Challenge on Post-K computer" (Challenge of Basic Science—Exploring Extremes through Multi-Physics and Multi-Scale Simulations). The present numerical simulations were tested on the Earth Simulator of the Japan Agency for Marine-Earth Science and Technology (JAMSTEC), while the final runs were performed on the K-computer installed at the Riken Advanced Institute for Computational Science.



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
