# Peer review of "Direct Lagrangian tracking simulation of droplet growth in vertically developing cloud"

_Atmospheric Chemistry and Physics, 2018_

## Referee Comment (RC1) · Anonymous Referee #1 · 4 May 2018

The model design and approach described in this paper are highly creative. The focus of the work is on Lagrangian aspects of precipitation formation through CCN activation, cloud droplet condensation growth, and ultimately coalescence growth. The number of published direct numerical simulations and Lagrangian-particle models is rapidly growing, but the key novelty of this work is the configuration of the simulation domain to have an extreme aspect ratio: 1 cm x 1 cm x 3 km for the main set of simulations. CCN, cloud droplets, and precipitation particles are discrete and tracked in three dimensions within the simulation domain, with condensation and coalescence growth directly calculated. The paper provides several insights into the coalescence growth process, which directly result from the adopted Lagrangian approach. I have suggested a few places

where some additional analysis may provide even further insight, although I realize that this is a first effort in that direction. I anticipate that this type of simulation approach will continue to yield insights into the transition from condensation to coalescence growth, statistical aspects of stochastic condensation, etc. I recommend publication, after the following suggestions have been considered.

Page 1, line 1: "all the warm-rain processes" may be a dangerous thing to say. For example, full hydrodynamic interactions are not represented. Perhaps better to state that "key warm-rain processes" are included.

Page 1, line 5: Euler framework should be Eulerian framework.

Abstract: the meaning of "surface raindrops" may not be clear in the abstract. Later in the paper it is explained, but at least in one place in the abstract, better to define as raindrops that reach the surface.

Page 1, line 21: it would be appropriate to include more citations of work taking a direct tracking Lagrangian approach, e.g., Vaillancourt et al., Wang et al., Kumar et al., Chen et al. J Atmos Sci 2018.

Page 2, line 21: transportation should be transport.

Section 2.2.1: specify the maximum height change, e.g., for the top of mixed layer initially at 740 m.

Equation 3: confusing to use notation F for acceleration. Perhaps f would be more consistent with typical notation.

Page 4, line 31: also should cite other DNS papers where hydrodynamic interactions have been shown to be important (e.g., Wang and colleagues).

Equation 7: the purpose of this stochastic initial radius is not clear. Why not simply grow cloud droplets from the activated CCN size?

Table 2: Specify that "Initial number concentration" refers to initial CCN concentration.

Page 8, line 17: should "homogeneously distributed" be "uniformly distributed"? Particles are not clustered, so I assume they are distributed with uniform probability.

Page 8, lines 23-24: here, reference could be made again to Table 1 so that the vertical gap can be properly understood.

Page 9, line 15: where is Table 4.1?

Page 10, lines 13-15: is there a "remarkable difference"? It is not clear to me what "sharp ridge" is referred to, so please explain more clearly.

Page 11, line 6: provide more explanation of what is meant by "Lagrangian statistics". For example, path history of collision times and sizes, etc.

Page 11, lines12-14: the observation that raindrops that reach the surface consist of CCN initially found below 900 m does not seem surprising, given that the model has no mechanism for entrainment. If I am missing something subtler, please explain.

Page 11, lines 19-20: This finding is intriguing. It could be interesting to see a pdf of droplet size at the time of first collision.

Page 11, line 22: need to define "Top-1" more clearly, e.g., the first droplet to reach the surface.

Figure 6: this is really interesting because it is clear that eventually the "top-1" drop collects other collector drops. It would be enlightening to see the transition from collection of cloud droplets to collection of collector drops (analogous to autoconversion versus accretion, perhaps). Although it would not show the time history, perhaps a pdf of droplet sizes that are collected by the "top-1" drop would be helpful. This is not required, I am just suggesting that there is more in the results that could be learned here. (Similar is true for the above comment on Page 11, lines 19-20.)

Page 14, lines 22-26: the supplementary movie is fantastic, definitely something I will show to students in the future. I recommend that a different term than "look-up view"

be used. Perhaps "upward-looking view" or "upward view"?

Page 15, lines 7 and 10: I think "statistics on" should be "statistics of".

Section 5: It might be useful to discuss similarities of this nearly-one-dimensional modeling approach to the one-dimensional turbulence models that have been used in cloud physics calculations (e.g., linear eddy modeling, Su et al Atmos. Res. 1998).

Section 5: It also would be useful to discuss how the simulation approach can be used for future studies of stochastic aspects of coalescence, which have been a topic of recent interest (e.g., Kostinski et al BAMS 2005, Bec et al Phys Rev E 2016, Wilkinson Phys Rev Lett 2016)
* * *

---

## Referee Comment (RC2) · Anonymous Referee #2 · 26 Jul 2018

**Report on "Direct Lagrangian tracking simulation of droplet growth in vertically developing cloud" (acp-2018-328) by Yuichi Kunishima and Ryo Onishi**

This paper reports the numerical study of evolution of the warm-rain process in a vertically developing cloud from microscopic view points. The simulation includes almost all processes such as cloud condensate nuclei (CCN) activation, condensational growth, collisional growth, and droplet gravitational settling, hydrodynamic interaction among the droplets. The obtained results are compared with the results of the conventional

spectral-bin simulations, and reasonable agreement is obtained. Particular feature of this study is to track the Lagrangian dynamics of all droplets within a extremely vertically elongated domain, almost one-dimensional domain. From the Lagrangian simulation the authors successfully obtained the history of evolution of the rain drops that reach the ground, as to how one rain drop is formed from many cloud droplets from the nucleation, condensation, collision-coalescence. The paper is well written and the results are very interesting and useful to the community. Therefore I would recommend the publication to the Atmos. Chem. Phys. when the point raised below is properly addressed.

1. The domain is almost one dimensional. The computational grid for the Navier-Stokes and transport equations is very elongated in the vertical direction. The aspect ratio is more than 12 from Table 2. The number of the grid points in the horizontal direction is only about 10, while more than 76,000 in the vertical direction. In this case, are the fluid variables properly solved? Is it necessary to solve these fluid equations? The fluids could simply be replaced by the one dimensional model.

2. In recent studies of micro physical processes, the importance of the turbulence is stressed. In the present simulation, no turbulence effects are taken into account. Could the authors comment on its effects on the results?

---

## Author Comment (AC1) · 3 Sep 2018

Thank you for your positive and insightful comments, recommending the acceptance of our manuscript for ACP. We answer your questions one by one below.

Q1: The domain is almost one dimensional. The computational grid for the Navier-Stokes and transport equations is very elongated in the vertical direction. The aspect ratio is more than 12 from Table 2. The number of the grid points in the horizontal direction is only about 10, while more than 76,000 in the vertical direction. In this case, are the fluid variables properly solved? Is it necessary to solve these fluid equations? The fluids could simply be replaced by the one dimensional model.

[Figure]

Answer: As described in subsection 2.2.1, the flow was prescribed as Eq. (1) in the present LCS simulation for KiD warm-1 case, which was designed to minimizes feedback between dynamics and microphysics. In case the flow had been solved explicitly, we would have had to be careful about the influence of the high aspect ratios of computational grids.

Q2: In recent studies of micro physical processes, the importance of the turbulence is stressed. In the present simulation, no turbulence effects are taken into account. Could the authors comment on its effects on the results?

Answer: We believe that the in-cloud turbulence enhances droplet growth in clouds. We will discuss the enhancement quantitatively as a next step.

———————————————

---

## Author Comment (AC2) · 4 Sep 2018

Reply to Referee#1
Thank you for reading our manuscript carefully and giving positive and insightful comments. We appreciate your understanding the novelty of the work and recommending its acceptance for ACP. We answer your comments and questions one by one below. As will be seen, we have added some discussion in the revised manuscript according to your insightful suggestion.

+————————————————

*Comment-1: Page 1, line 1: "all the warm-rain processes" may be a dangerous thing to say. For example, full hydrodynamic interactions are not represented. Perhaps better to state that "key warm-rain processes" are included.*

Answer:
We have reworded accordingly. We'd like to emphasize, for clarity, that the hydrodynamic interaction is represented in the present simulation. If you point out the lubrication force, yes you're right. The lubrication force is not properly represented in the present simulation. This may affect the initial collision growth rate to some extent.

+————————————————

*Comment-2: Page 1, line 5: Euler framework should be Eulerian framework.*

Answer:
Reworded 'Euler' to 'Eulerian' accordingly (not only in L5 in Page1, but many).

+————————————————

*Comment-3: Abstract: the meaning of "surface raindrops" may not be clear in the abstract. Later in the paper it is explained, but at least in one place in the abstract, better to define as raindrops that reach*

*the surface.*

Answer:

Corrected the corresponding part in the abstract as

"*the surface raindrops that reach the ground surface*" .

+--------------------------

*Comment-4: Page 1, line 21: it would be appropriate to include more citations of work taking a direct tracking Lagrangian approach, e.g., Vaillancourt et al., Wang et al., Kumar et al., Chen et al. J Atmos Sci 2018.*

Answer:

There is growing number of studies that compute the droplet growth with the direct tracking Lagrangian approach. It is not appropriate to include all of them, but we added some of them accordingly (Grabowski and Wang 2013, Kumar et al. 2013JAS, Chen et al. 2018JAS).

References

- Grabowski WW and Wang L-P, Growth of Cloud Droplets in a Turbulent Environment, Annu. Rev. Fluid Mech.. 45, 293-324 (2013).
- B. Kumar, J. Schumacher, R. A. Shaw, Lagrangian Mixing Dynamics at the Cloudy-Clear Air Interface, J. Atmos., Sci., 71, 2564-2580 (2014)
- S. Chen, M. K. Yau, and P. Bartello, Turbulence Effects of Collision Efficiency and Broadening of Droplet Size Distribution in Cumulus, J. Atmos., Sci., 75, 203-217 (2018)

+--------------------------

*Comment-5: Page 2, line 21: transportation should be transport.*

Answer:

Corrected accordingly (in conclusion as well).

+------------------------

*Comment-6: Section 2.2.1: specify the maximum height change, e.g., for the top of mixed layer initially at 740 m.*

Answer:
According to your comment, we have added the sentence after Eq. (1): *"This prescribed updraft lifts all the air parcels by ($\int wdt=$) 764m."*

+------------------------

*Comment-7: Equation 3: confusing to use notation F for acceleration. Perhaps f would be more consistent with typical notation.*

Answer:
We have changed the notation accordingly (bold **F** to bold **f_coll**). We have also changed notation f for non-linear drag coefficient to $\alpha$ to avoid a confusion.

+------------------------

*Comment-8: Page 4, line 31: also should cite other DNS papers where hydrodynamic interactions have been shown to be important (e.g., Wang and colleagues).*

Answer:
We cited Ayala et al. (2007) in Subsection 3.1 in the original manuscript. According to your comment, we have also cited Ayala et al. (2007) in the corresponding paragraph: *"...which is one of the so-called superposition methods (e.g., Ayala et al., 2007), ..."*

+------------------------

*Comment-9: Equation 7: the purpose of this stochastic initial radius is not clear. Why not simply grow cloud droplets from the activated CCN size?*

Answer:

*This stochastic procedure is introduced to avoid detail calculations of the CCN activation process, in which microscale supsersaturation fluctuations and CCN size fluctuations are to be properly represented. The present procedure reproduces realistic size distributions of activated droplets without those detail calculations.* We have added this description after Eq.7 in the revised manuscript.

+------------------------

*Comment-10: Table 2: Specify that "Initial number concentration" refers to initial CCN concentration.*

Answer:

We have revised it accordingly.

+------------------------

*Comment-11: Page 8, line 17: should "homogeneously distributed" be "uniformly distributed"? Particles are not clustered, so I assume they are distributed with uniform probability.*

Answer:

Revised accordingly. It meant as you conjectured.

+------------------------

*Comment-12: Page 8, lines 23-24: here, reference could be made again to Table 1 so that the vertical gap can be properly understood.*

Answer:
We have inserted a reference to Table 1 accordingly as "*...the domain top (z=3000m as in Table 1) and ...*"

+------------------------
*Comment-13: Page 9, line 15: where is Table 4.1?*

Answer:
We have corrected it to Table 3.

+------------------------
*Comment-14: Page 10, lines 13-15: is there a "remarkable difference"? It is not clear to me what "sharp ridge" is referred to, so please explain more clearly.*

Answer:
The "sharp ridge" is exhibited by the contour line for 1.5x10-3 kgm-3 in the LCS result. We have added the contour values in Fig.3 and inserted the guidance as "*The LCS result shows a sharp ridge, drawn by the contour line for 1.5x10$^{-3}$ kgm$^{-3}$, inside the cloud layer, ...*"

+------------------------
*Comment-15: Page 11, line 6: provide more explanation of what is meant by "Lagrangian statistics". For example, path history of collision times and sizes, etc.*

Answer:
"Lagrangian statistics" mean the statistics on histories of individual Lagrangian particles such as the statistics on path histories of collision times of individual rain drops and those on the growth histories of individual cloud droplets. We have added an explanation in the corresponding sentence as "*...they can provide Lagrangian*

*statistics, on histories of individual Lagrangian particles."*

+—————————————————

*Comment-16: Page 11, lines12-14: the observation that raindrops that reach the surface consist of CCN initially found below 900 m does not seem surprising, given that the model has no mechanism for entrainment. If I am missing something subtler, please explain.*

Answer:

*The air parcel was lifted by 764m, while the cloud top was about 2,000m as in Fig.3. The CCN initially located below 1,200m (~2,000-764 m) in height participated the cloud layer formation but those between 900 and 1,200m did not participate the formation of the surface raindrops. We have added this in the revised manuscript.*

+—————————————————

*Comment-17: Page 11, lines 19-20: This finding is intriguing. It could be interesting to see a pdf of droplet size at the time of first collision.*

Answer:

Let us start with pointing out that the mean value of the suggested pdf is not directly related with the value 36.1um in the corresponding part. The mean value will be much smaller than 36.1um since the first collision seems to happen before 300s (see Fig.7=Fig.6 in the original manuscript) while condensation growth continues afterwards until 600s when the updraft is ceased. We are, however, interested in the suggested pdf regarding droplet size at the time of first collision of surface raindrops, as well as the pdf of the time itself. The pdf of sizes of collected droplets will be interesting as well. For the present paper, we have added one pdf graph (new Fig.6), but we leave further pdfs in a future work.

+------------------------

*Comment-18: Page 11, line 22: need to define "Top-1" more clearly,
e.g., the first droplet to reach the surface.*

Answer:
The Top-1 surface raindrop is the raindrop that reached the ground
surface firstly in each run. We have added this explain as *"... the Top-
1 surface raindrop, which reached the ground surface firstly in each
run."*

+------------------------

*Comment-19: Figure 6: this is really interesting because it is clear
that eventually the "top-1" drop collects other collector drops. It
would be enlightening to see the transition from collection of cloud
droplets to collection of collector drops (analogous to autoconversion
versus accretion, perhaps). Although it would not show the time history,
perhaps a pdf of droplet sizes that are collected by the "top-1" drop
would be helpful. This is not required, I am just suggesting that there
is more in the results that could be learned here. (Similar is true for
the above comment on Page 11, lines 19-20.)*

Answer:
Thanks for the insightful comment. Yes, the pdf of droplet sizes that
are collected by surface raindrops is definitely of interest. As
commented in the reply above, we will show pdfs in more detail in a
future work.

+------------------------

*Comment-20: Page 14, lines 22-26: the supplementary movie is fantastic,
definitely something I will show to students in the future. I recommend
that a different term than "look-up view" be used. Perhaps "upward-*

*looking view" or "upward view" ?*

Answer:
We have adopted "upward-looking view" for the revised supplement movie.

+—————————————
*Comment-21: Page 15, lines 7 and 10: I think "statistics on" should be "statistics of".*

Answer:
We have corrected them accordingly.

+—————————————
*Comment-22: Section 5: It might be useful to discuss similarities of this nearly-one-dimensional modeling approach to the one-dimensional turbulence models that have been used in cloud physics calculations (e.g., linear eddy modeling, Su et al Atmos. Res. 1998).*

Answer:
The linear-eddy modeling approach is an interesting way of simulating the mixing&entrainment and condensational growth of cloud droplets. One clear drawback of the present nearly-one-dimensional approach is that it cannot consider the large-scale entrainment. If integrated with the linear-eddy modeling approach, the drawback may be overcome.

+—————————————
*Comment-23: Section 5: It also would be useful to discuss how the simulation approach can be used for future studies of stochastic aspects of coalescence, which have been a topic of recent interest (e.g., Kostinski et al BAMS 2005, Bec et al Phys Rev E 2016, Wilkinson Phys Rev Lett 2016)*

Answer:

We are also interested in the stochastic aspects of droplet growth in clouds. Previous theoretical and numerical studies discussed the stochasticity or luckiness in possible rapid onset of surface precipitations under idealized conditions, specifically, adopting a parcel (zero-dimensional) framework and ignoring condensation/evaporation and gravitational settlings. For example, Kostinski & Shaw (2005)BAMS and Wilkinson (2016)PRL discussed the stochastic aspect of collision growth ignoring the condensation growth in cloud parcels. They showed possible large deviations in the time required for droplets to grow from 10 to 50 um in radius. However, their conclusions cannot be simply applied to the real world since the process of droplet falling through unsaturated air below cloud layers is ignored. The air below the cloud layers is unsaturated (particularly for the onset time of surface precipitations) and the unsaturated air acts as a barrier that makes it difficult for rain drops to pass through via evaporating them. A rain drop must grow large enough to overcome this evaporation barrier in order to become a surface raindrop. Figure 6 (newly added in the revised manuscript) actually shows that the rain drops smaller than 200um in radius at cloud bottom (i.e., z=600m) evaporated completely before reaching the ground surface. This clearly suggests that the time required to grow from 10 to 200um, rather than to 50um, in radius should be measured for the discussion of the rapid onset of surface precipitations. The present quasi-1D approach can provide a practical platform for more realistic discussion than the classical parcel (zero-dimensional) approach. We have added Fig.6 and the information above in the revised manuscript.

 "*Figure 6 shows the probability density function (PDF) of the surface raindrops when they reach at 600m height. Raindrops smaller than 200um in radius evaporated completely in the unsaturated air below the cloud bottom (z~600m) before reaching the ground surface, and thus could not become surface raindrops. That is, the unsaturated air near the surface*

*acts as a barrier that prevents small raindrops from reaching the ground surface. This process cannot be represented in the parcel (zero-dimensional) concept and it is thus often neglected in theoretical works. For example, Kostinski & Shaw (2005) and Wilkinson (2016) discussed, neglecting that process, the stochastic aspect of the onset of surface precipitations and showed possible large deviations in the time required for droplets to grow, via collisions, from 10 to 50 um in radius. The present result, however, suggests that the time required to grow from 10 to 200um, rather than to 50um, in radius should be measured for the discussion of the rapid onset of surface precipitations."*

The referee also suggested another paper by Bec et al. (2016)PRL. It discussed under a droplet volume fraction of $0.5 \times 10^{-4}$, which is 10 times denser than that in the typical clouds. A tenfold denser condition leads to 100 times more frequent collisions, i.e., to 1/100 of collision intervals. Such short intervals would lead to much larger (i.e., unrealistically large) correlations between successive collisions than actual correlations in clouds, and make it difficult to estimate realistic stochasticity. It should be emphasized that the present simulation led to realistic volume fractions of $O(10^{-6})$ (Note that the LWC of $O(10^{-3})$ kg/m3 as in Fig.3 and droplet density of $O(10^3)$kg/m3.).

References

● Kostinski, A. B. and Shaw, R. A.: Fluctuations and Luck in Droplet Growth by Coalescence, Bulletin of the American Meteorological Society, 86, 235-244 (2005)
● Wilkinson, M.: Large Deviation Analysis of Rapid Onset of Rain Showers, Phsical Review Letters, 116, 018 501 (2016)